# A New Strategy for the Synthesis of Hydroxyl Terminated Polystyrene-*b*-Polybutadiene-*b*-Polystyrene Triblock Copolymer with High Cis-1, 4 Content

**DOI:** 10.3390/polym11040598

**Published:** 2019-04-02

**Authors:** Xin Min, Xiaodong Fan

**Affiliations:** Department of Applied Chemistry, School of Nature and Applied Sciences, Northwestern Polytechnical University, Dongxiang Road 1, Xi’an 710072, China; 15991672082@163.com

**Keywords:** hydroxyl terminated polystyrene-*b*-polybutadiene-*b*-polystyrene triblock copolymer, high cis-1,4 content, nickel catalyst

## Abstract

This work reports the preparation of a hydroxyl terminated polystyrene-*b*-polybutadiene-*b*-polystyrene triblock copolymer (SBS) with high cis-1, 4 content via a novel nickel catalyst, [η^3^-Ni(CH_2_CHCHCH_2_OOCH_3_)][BPh^F^_4_]. FT-IR, ^1^H-NMR, and ^13^C NMR indicated that the polybutadiene segment of the copolymer contains greater than 90% cis-1, 4 structure, indicating achievement of the objective. Toward the functionalization goal, a hydroxyl group was successfully introduced at the end of the triblock copolymer (HO–SBS–OH). The results of gel permeation chromatography (GPC) revealed that the polymer is indeed a triblock copolymer, with no traces of homopolymer. Differential scanning calorimetry (DSC) showed that HO–SBS–OH synthesized using the novel catalyst had a lower glass transition temperature (*T_g_*) than HO–SBS–OH synthesized with an alkyl lithium catalyst. Therefore, the polymer synthesized via the novel catalyst contains high cis-1,4 content and displays excellent low-temperature mechanical properties.

## 1. Introduction

SBS copolymers have a wide range of applications as thermoplastic elastomers. However, traditional SBS copolymers have no functional groups meaning that they cannot be chemically modified or cross-linked to obtain alternating embedded copolymers. Therefore, there is a real interest in the development of an SBS block copolymer exhibiting functional groups, particularly hydroxyl groups, which will enable post-synthetic modification of the copolymer.

Currently, anionic polymerization is the most common method for the preparation of SBS copolymers [1]. However, this approach presents some disadvantages. For instance, a difference in the activity of butadiene and styrene monomers in polar solvents (e.g., THF) is necessary to guarantee the conversion of the monomers. Moreover, the microstructure of the resulting polymer is less controllable, especially for the polymerization of butadiene. Polybutadiene with low cis-1,4 content (35–40%) is obtained [2], resulting in a high *T_g_* and thus poor mechanical properties at low temperatures. This is mainly due to limited coordination between the alkyl lithium initiator and the olefin monomers. Thus, the chain end growth cannot be strictly controlled, yielding low cis-1,4 content in the polybutadiene blocks. The use of polar solvents causes additional problems [3,4,5,6,7,8], resulting in a polymer that exhibits poor mechanical properties at low temperatures. Due to the low barrier to the internal rotation, the *T_g_* of segments with cis-1,4 units is much lower than that of 1,2 unit-based structures (*T_g_* cis-1,4 = −106 °C, *T_g_* 1,2 = −15 °C). This means that the segment of cis-1,4 units can maintain flexibility at an extremely low temperature compared to the 1,2 unit-based structures. Moreover, the trans-1,4 unit can easily crystallize at low temperatures due to the alignment of chain segments; therefore, both 1,2 and trans-1,4 unit contents can seriously affect the low-temperature mechanical properties of polymers. Thus, the analysis of the extent of cis-1,4 content is a very useful parameter to judge the low-temperature flexibility of SBS copolymers. Based on these parameters, the development of a new method for the synthesis of HO–SBS–OH with high cis-1,4 content is necessary, though challenging.

While the synthesis of polystyrene-polybutadiene block copolymers with high cis-1,4 content via transition metal catalysts is highly studied [9,10,11,12,13,14,15,16,17,18,19,20,21,22,23,24,25,26,27,28], to the best of our knowledge, no report on the synthesis of hydroxyl terminated polystyrene-polybutadiene block copolymers in the presence of these catalysts is known. This is due to the fact that the surface of these catalysts requires suitable chemical modification to improve their activity in the presence of alkyl aluminum (e.g., methylaluminoxane and triethylaluminum) as co-catalysts. However, these co-catalysts lack hydroxyl groups, making them incompatible for the synthesis of hydroxyl terminated SBS copolymers (Scheme 1). Moreover, the synthesis of alkyl aluminum containing hydroxyl groups is challenging and may result in the failure of the alkylation reaction.

The ideal catalyst for the preparation of hydroxyl terminated SBS triblock copolymers with high cis-1,4 content should display three characteristics, as follows. (i) The catalyst should have available hydroxyl or protected hydroxyl groups on its surface. (ii) The catalyst must be highly active so that it does not require a cocatalyst to initiate styrene and butadiene polymerization. (iii) Transition metal-based catalysts should be used to ensure a high cis-1,4 content. Our previous work reported a new type of allyl nickel catalyst, [η^3^-Ni(CH_2_CHCHCH_2_OOCH_3_)][BPh^F^_4_] [29], which meets all three criteria. Thereby, this catalyst fits the current needs in that (a) there are protected hydroxyl groups on the catalyst surface, (b) [η^3^-Ni(CH_2_CHCHCH_2_OOCH_3_)][BPh^F^_4_] it can initiate styrene and butadiene polymerization without a cocatalyst, and (c) nickel, which exhibits strong coordination to olefin monomers, is the center of the catalyst to ensure the high cis-1,4 content of the polybutadiene sections.

## 2. Materials and Methods 

### 2.1. Materials

Bis(1,5-cyclooctadiene)nickel(0) (96%, Aladdin, Shanghai, China), 2-Butene-1,4-dioldiacetate, (99%, Aladdin), butadiene (15 wt % in toluene, TCI Chemicals, Shanghai, China), styrene (AR, Acros, Shanghai, China), 3-iodopropanol (99%, Aladdin), sodium methylate (30% wt % in methanol, Aladdin), diethyl ether (AR, Aladdin), n-hexane (AR, Aladdin), tetrahydrofuran (THF, AR, Aladdin), sodium tetrakis[3,5-bis(trifluoromethyl)phenyl]borate (99%, Aladdin) were used.

### 2.2. General Methods

All solvents and the styrene monomer were purified by distillation in the presence of calcium hydride and then stored with molecular sieves.

The molecular weight and molecular weight distribution of polymers were measured by gel permeation chromatography (GPC) using DAWN EOS type gel osmotic chromatography-polygon laser light scattering (SEC-MALL)(Santa Barbara, CA, USA). The eluent used THF; polymer concentration in the eluent was 1mg/ml; flowrate was 0.5 mL/min; temperature was 25 °C; *d*_n_/*d*_c_= 0.16. For the tri-block copolymer structure analyzed, the crude product of a tri-block copolymer was washed by acetone and n-hexane to remove homogeneous polystyrene and homogeneous polybutadiene, and then the product was refined repeatedly by fractional precipitation through mixed solvent of acetone and n-hexane until the GPC spectrum was unimodal, which can guarantee the purity of tri-block copolymer. Then the refined product was diluted in several gradient concentrations to measure the *d*_n_/*d*_c_ of the tri-block copolymer. Finally, molecular weight data can be measured. The ^1^H-NMR and ^13^C NMR analyses were performed on a Bruker Avance-400 NMR (Madison, WI, USA). The solvent used was CDCl_3_, and the interior label was tetramethylsilane (TMS). Instrument conditions: Magnetic field intensity was 9.4*T*; frequency was 400MHz; scan time was 16; work temperature was 25 °C. A Mettler-Toledo STARe equipment system (Zurich, Switzerland) was used to evaluate the glass transition of the polymer using differential scanning calorimetry (DSC)(Netzsch Instruments, Selb, Germany). Procedure: The first stage was room temperature–200 °C, and the heating rate was 20 °C/min kept at 200 °C for 10 mins; second stage was 200–130 °C, at a cooling rate of 10 °C/min, kept at −130 °C for 20mins; the third stage was −130–200 °C at a heating rate of 5 °C/min.

### 2.3. The Measuring Method of the Hydroxyl Value

The specific experimental method was as follows: “phthalic anhydride 35 g was dissolved in 250 mL pyridine (called acetylated mixture solution). After 24 h, samples were accurately weighed, and 8 mL acetylated mixed solution was added, then reflexing at 115 °C for 1h, then 8 mL pyridine and 16 mL water were added from the top of the condensation tube and continued to react for 15 min. Next, titration was operated by KOH solution (0.5mol/L in water), and phenolphthalein was the indicator. Hydroxyl value = (0.5(*V*_1_ − *V*_2_)*M*_n_)/1000 m, *V*_1_ = titrant’s volume of the blank sample, *V*_2_ = titrant’s volume of the experimental sample, *m* = the weight of the experimental sample.”, *M*_n_ = measured molecular weight by SEC.

### 2.4. Synthesis of [η^3^-Ni(CH_2_CHCHCH_2_OOCH_3_)][BPh^F^_4_]

Bis(1,5-cyclooctadiene)nickel(0) (55 mg) was added to a dry flask under an argon atmosphere and then diethyl ether (1 mL) was added slowly under an argon atmosphere as well. After cooling the system to 0 °C, 2-butene-1,4-dioldiacetate (34 mg, dissolved in 2 mL diethyl ether) was added dropwise. The solution was stirred at 0 °C for 30 min and the color gradually changed from yellow to red. Then sodium tetrakis[3,5-bis(trifluoromethyl)phenyl]borate (177.2 mg) was added to the system at 0 °C and the solution was stirred for another 30 min. Finally, the diethyl ether was removed under a vacuum and the mixture was washed with n-hexane under an argon atmosphere. A yellow–orange solid was obtained with a yield of 53%.

### 2.5. Synthesis of HO-SBS-OH with High cis-1,4 Content

The synthesis was performed according to the Schlenk technique. For the first polymerization step, 1 mL of styrene was solved in 4 mL of toluene under an argon atmosphere. Then, the catalyst ([η^3^-Ni(CH_2_CHCHCH_2_OOCH_3_)][BPh^F^_4_], 133.1 mg) was added to the solution, which became deep red, offering direct evidence that the polymerization started. The solution was stirred for 2 h at 50 °C. Then, the second polymerization step started, and 10 mL of butadiene solution (15 wt % in toluene) was added in the Schlenk flask and stirred at room temperature for 4 h. Meanwhile, the color of the solution turned from deep red to yellow. Afterward, the third polymerization step was begun, and 1 mL of styrene was added, and the yellow solution became deep red again. The solution was kept under stirring for another 2 h. Then, 0.2 mL of 3-iodopropanol was added and stirred for 1 h. To stop the reaction, 0.5 mL of methanol was added to the flask. The solvent was removed under a vacuum. The resulted polymer was dissolved in THF and washed 3 times with methanol. A colorless and transparent viscous liquid with a yield of 98% was obtained. The gravimetry method was used to determine the monomer conversions after each step.

The dry product abovementioned (1 g) was dissolved in 2 mL of THF and then 1 g of sodium methylate was added. The mixture was stirred at 50 °C for 2 h. After that, the solvent was removed under a vacuum, whereas the HTPB was re-dissolved in THF and washed 3 times with methanol. Finally, a colorless and transparent viscous liquid (HO–SBS–OH) was obtained with a yield of 99%. The gravimetry method was used to determine the monomer conversions after hydrolysis.

For tracking the progress of triblock copolymer, samples were taken out from the first polymerization step at set intervals and were washed 3 times with methanol. GPC could not be used to measure the molecular weight information of the first polymerization step until the weight of samples were constant. The GPC samples of the second polymerization step and third polymerization step should be treated with the same method as the first polymerization step. 

### 2.6. The Calculation Method of the Microstructure Content of Copolymer

The content of 1,4 units (cis and trans) and the 1,2 unit in copolymer were calculated from ^1^H-NMR data and using Equation (1). The results were double checked using FT-IR and quantified by applying Equation (2).
(1)$$B\%=(1-\frac{2I_{\left(5.01∼4.8\right)}}{2I_{\left(5.38\right)}+I_{\left(5.01∼4.8\right)}})\times 100\%$$
*I* = the integral area of ^1^H-NMR of the copolymer at the corresponding chemical shift,*B*% = the content of 1,4 structure in copolymer.
(2)$$\begin{array}{l} C_{C}\%=\frac{D_{\left(724{\mathrm{cm}}^{-1}\right)}/K_{c}}{D_{\left(724{\mathrm{cm}}^{-1}\right)}/K_{c}+D_{\left(911{\mathrm{cm}}^{-1}\right)}/K_{v}+D_{\left(967{\mathrm{cm}}^{-1}\right)}/K_{t}}\times 100\%\\ C_{V}\%=\frac{D_{\left(911{\mathrm{cm}}^{-1}\right)}/K_{v}}{D_{\left(724{\mathrm{cm}}^{-1}\right)}/K_{c}+D_{\left(911{\mathrm{cm}}^{-1}\right)}/K_{v}+D_{\left(967{\mathrm{cm}}^{-1}\right)}/K_{t}}\times 100\%\\ C_{t}\%=\frac{D_{\left(967{\mathrm{cm}}^{-1}\right)}/K_{t}}{D_{\left(724{\mathrm{cm}}^{-1}\right)}/K_{c}+D_{\left(911{\mathrm{cm}}^{-1}\right)}/K_{v}+D_{\left(967{\mathrm{cm}}^{-1}\right)}/K_{t}}\times 100\% \end{array}$$
*D* = Absorbance of the corresponding band in the FTIR spectrum, *K_c_* = Absorption coefficient of cis-1, 4 unit (720–740 cm^−1^), *K_v_* = Absorption coefficient of 1, 2 unit (911 cm^−1^), *K_t_* = Absorption coefficient of trans-1, 4 unit (967 cm^−1^), *C_c_*%, *C_v_*%, *C_t_*% = the content of cis-1, 4 unit, 1, 2 unit, and trans-1,4 unit, respectively.


The calculation method of microstructure content via ^13^C NMR was performed according to the reference [16,27].

## 3. Results and Discussion

### Synthesis of [η^3^-Ni(CH_2_CHCHCH_2_OOCH_3_)][BPh^F^_4_]

The synthesis of [η^3^-Ni(CH_2_CHCHCH_2_OOCH_3_)][BPh^F^_4_] is shown in Scheme 2. Details on the characterization of [η^3^-Ni(CH_2_CHCHCH_2_OOCH_3_)][BPh^F^_4_] are found in the references [29].

Scheme 3 displays the synthesis of HO–SBS–OH with high cis-1,4 content. SBS triblock copolymers with protected hydroxyl groups were obtained by a one-pot method. The monomers were added in a stepwise fashion, whereas 3-iodopropanol was used to install the additional terminal hydroxyl group. Finally, HO–SBS–OH was obtained after hydrolysis.

The ^1^H-NMR spectrum of HO-SBS-OH is shown in Figure 1. H_1_ corresponds to the characteristic chemical shift of 1,4-polymerized butadiene while H_3_ denotes the characteristic chemical shift of the hydrogens in the benzene ring of polystyrene. H_4_ is the chemical shift of the hydrogens of the CH_2_ unit connected to the hydroxyl functionality. The identification of the chemical shifts attributed to H_1_ and H_3_ indicates that the polymer product is indeed the polystyrene–polybutadiene block copolymer, whereas the presence of H_4_ indicates successful incorporation of the hydroxyl group in the structure. To confirm that the ester-protected hydroxyl group was incorporated by the initiator, FT-IR was used to analyze the copolymer before hydrolysis (Figure 2). A band at 1700 cm^−1^, attributed to a C=O bond, was identified in the spectrum, indicating the presence of an ester bond in the polymer. The band at 3500 cm^−1^ is characteristic of hydroxyl groups, indicating that they were successfully incorporated into the polymer. Moreover, the bands identified at 726, 967, and 911 cm^−1^ are typical for cis-1,4-, trans-1,4-, and 1,2-polybutadiene, respectively, while the band at 697 cm^−1^ is characteristic to the benzene ring of polystyrene. Based on a literature review [15], it can be confirmed that the SBS block copolymer exhibiting terminal hydroxyl groups has been successfully obtained.

Scheme 3 suggests that the synthetic sequence proceeds in the following order: Polymerization→addition of iodopropanol→hydrolysis. Given this information, the polymer exists in three forms dependent on the status of the synthesis: Prior to the addition of iodopropanol, after the addition of iodopropanol (prior to hydrolysis), and after hydrolysis. To prove the validity of our proposed reaction sequence, we examined the three forms of the polymer via FT-IR (Figure 3). The hydroxyl peak appeared after adding iodopropanol, which suggests that iodopropanol reacts with the active end of the polymer, thereby installing a terminal hydroxyl group. The absorption peak corresponding to C=O, which results from the catalyst, disappeared from the FT-IR spectra after hydrolysis, suggesting successful hydrolysis from ester to hydroxyl functionality. 

In addition, the HO–SBS–OH block copolymer was characterized by ^13^C NMR to assess the extent of cis-1,4 content (Figure 4). The spectrum displays a peak at 27.4 ppm, which is characteristic of the CH_2_ of a cis-1,4-butadiene unit [15] and another at 32.7 ppm, typical for trans-1,4 carbons. Based on the integration of the peaks, cis-1,4 content was calculated at 90.4% for the HO–SBS–OH block copolymer synthesized via the novel [η^3^-Ni(CH_2_CHCHCH_2_OOCH_3_)][BPh^F^_4_] catalyst. This is significantly higher than the value obtained for the same polymer synthesized by anionic polymerization (38%). Moreover, the *M*_n_ calculated by ^1^H and ^13^C NMR was 21352 g·mol^−1^, which coincides with the value obtained by size exclusion chromatography (SEC) (21896 g·mol^−1^). The integral data and computational processes are available in the Appendix A.

Table 1 displays the characteristics of HO-SBS-OH synthesized via [η^3^-Ni(CH_2_CHCHCH_2_OOCH_3_)][BPh^F^_4_]. The polystyrene content in the copolymer is consistent with the molar ratio of the monomers. Similarly, the molecular weight of the copolymer is consistent with the molar ratio between the monomers and initiator. The molecular weight distribution is less than 1.2, which suggests that the process is a living polymerization. The hydroxyl value suggests that there is an average of 1.97–2.01 hydroxyl functionalities in each macromolecule, which supports the conclusion that di-hydroxyl terminated polymers were synthesized.

GPC was used to track the polymerization process. After purifying with n-hexane and acetone, the experimental results illustrated in Figure 5a show a single peak for polystyrene-*b*-polybutadiene diblock copolymer (PSB) (the ^1^H-NMR of St polymerization process with different polymerization time is shown in Appendix A), which indicates that there is no homopolymer in the diblock copolymers. Similarly, the GPC traces for the SBS triblock copolymer also display a single peak, which indicates high purity of the triblock copolymer without traces of diblock copolymer. Table 2 shows the detailed molecular weight data of each polymerization step.

After purifying with n-hexane and acetone, the surface of HO–SBS–OH triblock polymer was investigated by atomic force microscopy AFM (Figure 5b). The results of this analysis confirm the absence of the homopolymer. Moreover, HO–SBS–OH shows typical microphase separation. These results strengthen the conclusion that the HO–SBS–OH block copolymer was successfully synthesized with the novel [η^3^-Ni(CH_2_CHCHCH_2_OOCH_3_)][BPh^F^_4_] catalyst.

The glass transition temperature of the SBS copolymer was evaluated by DSC. To highlight the improvements induced by the high content of cis-1,4 units, the data of the polymer synthesized with the novel Ni catalyst (Ni/HO–SBS–OH) are compared with those obtained for the polymer synthesized by anionic polymerization (Li/HO–SBS–OH) [30]. These results are summarized in Figure 6. ^1^H and ^13^C NMR (shown in Appendix A) show that the cis-1,4 content of Li/HO–SBS–OH is 39.6%. As can be seen, there is no significant difference in the *T_g_* of the polystyrene segment while the *T_g_* of the polybutadiene segment in Ni/HO–SBS–OH was significantly lower than the corresponding segment in Li/HO–SBS–OH, indicating that Ni/HO–SBS–OH has an enhanced temperature performance as compared to Li/HO–SBS–OH.

## 4. Conclusions

In summary, this work proposed a new strategy to synthesize a hydroxyl terminated polystyrene-*b*-polybutadiene-*b*-polystyrene triblock copolymer (HO–SBS–OH) with high cis-1,4 content by using [η^3^-Ni(CH_2_CHCHCH_2_OOCH_3_)][BPh^F^_4_]. FT-IR and ^1^H-NMR showed that the hydroxyl group was successfully introduced in the structure of the copolymer. ^13^C NMR results revealed that the cis-1,4 content of HO–SBS–OH is 90.4%, which is significantly higher than the value obtained by anionic polymerization (38%). The results of GPC and AFM proved that the obtained material is a pure block copolymer without traces homopolymer. DSC analysis evidenced that the *T_g_* of the polybutadiene segment of Ni/HO–SBS–OH was significantly lower than that of Li/HO–SBS–OH, which greatly enhanced the mechanical properties of the resulting copolymer.

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
