# Peer review of "A New Strategy for the Synthesis of Hydroxyl Terminated Polystyrene-*b*-Polybutadiene-*b*-Polystyrene Triblock Copolymer with High Cis-1, 4 Content"

_polymers, 2019, doi:10.3390/polym11040598_

Round 1

Reviewer 1 Report

The comments made in the previous review have been completely addressed.  Thus, I recommend acceptance.  The writing/grammar still needs improvement throughout the manuscript, as does the formatting of the figures. (Figures 2a/2b should be presented as a single figure with one caption; caption on Scheme 1 is not well written; superscripts are needed on MW units in Table 1 and 2; improper spacing around tables and figures.) In addition, line spacing is not uniform among paragraphs.

Lines 224-225 and 229-230 states that the absence of homopolymer was confirmed. It needs to be emphasized that this is due to purification steps taken, as homopolymer was formed during the synthesis steps.

Author Response

Thank you very much for your suggestions. We have sough help of two native English professors to check and improve the writing and grammar.

We revised figure 2(a) to figure 2 and figure 2(b) to figure 3 and the other figure numbers was respectively changed.

To clearly illustrating Scheme 1. We revised caption of Scheme 1 to “Scheme 1 the reason why existing transition metal catalyst can not synthesized functional groups terminated polystyrene-polybutadiene block copolymers with high cis-1,4 content”.

We revised the format and superscripts of Table 1 and Table 2 which is highlighted in main text, moreover, we uniform the line spacing of paragraphs . For emphasizing, we revised the sentence “The experimental results illustrated in figures 4(a) and (b) show a single peak for polystyrene-b-polybutadiene diblock copolymer (PSB), which indicates that there is no homopolymer in the diblock copolymers.” to “After purifying with n-hexane and acetone, the experimental results illustrated in figures 5(a) show a single peak for polystyrene-b-polybutadiene diblock copolymer (PSB), which indicates that there is no homopolymer in the diblock copolymers.”. Moreover, we revised the sentence “The surface of HO-SBS-OH triblock polymer was investigated by AFM.” to “After purifying with n-hexane and acetone, the surface of HO-SBS-OH triblock polymer was investigated by AFM (figure 5(b)).”.

Reviewer 2 Report

The authors politely answered all my questions and comments. Therefore, the manuscript will be suitable for this journal.

Author Response

R: Thank you very much for your comments to the manuscript. We feel honoured that you are satisfied with our work.

This manuscript is a resubmission of an earlier submission. The following is a list of the peer review reports and author responses from that submission.

Round 1

Reviewer 1 Report

In this paper, hydroxyl terminated poly(styrene)-b-poly(butadiene)-b-poly(styrene) triblock copolymers were prepared using a polymerization method with nickel catalyst. The triblock copolymers had the cis-1,4 structure with more than 90% content, and the molecular weights were controlled by the polymerization conditions. The physicochemical properties of triblock copolymers were investigated using FT-IR, NMR, DSC, and AFM. The paper shows important results about the hydroxyl terminated poly(styrene)-b-poly(butadiene)-b-poly(styrene) triblock copolymers with high contents of cis-1,4 structures. However, the analysis of terminated hydroxy groups was not enough. And DSC results lack the discussion part about the relationship between the cis-1,4 structures and physicochemical properties. Moreover, there are errors including references, chemical structures, and spelling errors. Therefore, the manuscript will be suitable for this journal of ‘Polymers’ after major revision. My questions and comments are shown as follow.

Q1. The analysis of terminated hydroxyl groups was not enough. Did all chain ends turn to hydroxyl groups? You should add the conversion ratio of OH.

Q2. The relationship between the cis-1,4 structure and physicochemical properties is not clear. In Figure 5, you should add the DSC data of triblock copolymers with low content of cis-1,4 structure.

Q3. There are mistakes including references, chemical structures, and spelling errors. (e.g.

University name: P.1L.6 Northwster, Northwestern; P.2 L.4, highly studied [4-28]: you have to check the reference numbers; in Figure 2, the structure of the triblock copolymer may be error. 

Reviewer 2 Report

The objectives of the study are clearly outlined. The results, especially that shown in Figure 5, indicates that the authors successfully synthesized an SBS structure that contains higher cis content. Thus, it is a nice contribution. However, the manuscript is missing key details. Major revision and re-review are need, to re-evaluate the contribution after the descriptions of polymer characterization, polymer synthesis, polymer modification procedures, and some characterization results, are expanded, as outlined below:

1.      Polymer characterization (missing details line 79-85):

-        NMR: solvent used, peak assignments, instrument conditions

-        SEC-MALL: eluent used, polymer concentration in eluent, flowrate and temperature, calibration strategy. In particular, how is a tri-block structure analyzed, as poly(styrene) and poly(butadiene) have different dn/dc values.

-        DSC procedure (heating rate, etc…) needs to be specified.

2.      Polymer synthesis (line 95-110):

-        ST was polymerized for 2 h at 50 C, followed by butadiene for 4 h at room temperature, followed by ST for 2 h at room temperature. What is the expected polymer structure, assuming full monomer conversions (i.e., values of x,y,z in poly(ST_x-block-Butadiene_y-block-ST_z).

-        Do you know if full conversion was reached at the end of each polymerization stage? How was this checked? Overall yield is reported as 99%, but if ST was not completely converted in the first stage, your polymer may contain some random ST-Butadiene copolymer linkages.

-        Figure 4 indicates that samples were taken after each polymerization step. This sampling needs to be described in the experimental section. Was gravimetry or NMR used to determine the monomer conversions after each step? Details need to be provided.

3.      Polymer modification (attachment of hydroxyl functionality):

-        The synthesis pathway is shown in Scheme 3. However, no reaction details are provided for the reaction with iodopropanol or for the hydrolysis step (mentioned at line 135-136). Full experimental details need to be provided.

These experimental details need to be provided before the manuscript can be accepted for publication.

In addition, the following details need to be added to the discussion/results section:

4.      It is customary to report the block lengths (both expected and measured) when describing a block terpolymer such as the SBS synthesized here. This needs to be specified!

5.      What is the overall ratio of styrene to butadiene to OH end groups determined from the NMR analysis (Figure 1)? Does the calculated MW from NMR agree with that measured by SEC? Are Table 1 Mn values from NMR or SEC?

6.      The NMR spectra (Figure 1) shows the presence of OH groups. It does not, however, prove that the chains are di-functional (HO-SBS-OH), as claimed. Has any experimental work been done (such as crosslinking experiments) to prove that the chains are di-functional?

7.      Figure 4 plots the raw output from the SEC analysis. However, absolute values of Mn are reported in Table 1, and MALLS detection was used. I suggst that this figure should be re-worked to show the MWDs after the raw data are processed. At the very least, it is necessary to report the Mn values measured after each block is added to the polymer chain (PS to PSB to PSBS). This data should be reported for each experiment included in Table 1.

8.      There seems to be significant overlap in the three SEC traces for at the low MW side (time=17 min). This would suggest that some dead polymer chains are formed (homopolymer or di-block copolymers). Can the authors comment on this observation? Reporting the dispersities and Mn values, as requested with the point above, will be important for this discussion.

Other points:

9.      Language (grammar, sentence structure) should be improved. (For example, the phrase “indicating the well achievement of the objective” in the abstract.)

10.   Figure and scheme captions need to be expanded to fully describe what is being shown.

11.   Abstract: First sentence defines styrene-butadiene-styrene block copolymer as (HO-SBS-OH). But it is not until later in the abstract that it is mentioned that hydroxyl groups are added.  It would be better to refer to the original triblock polymer (before modification) as SBS.

12.   Line 33: should be “anionic polymerization”

13.   Line 39-40: It is claimed that the reason behind low cis-1,4 content is due to coordination of the alkyl-Li initiator. References should be provided to support this statement.